# OpenReview forum: "Strategic Obfuscation of Deceptive Reasoning in Language Models"
_ICLR.cc/2026/Conference — ICLR 2026 Poster_

### Official Review · Reviewer_Hzkp · 2025-10-18

**Soundness:** 2
**Presentation:** 3
**Contribution:** 2
**Rating:** 4
**Confidence:** 4

**Summary:**

The paper describes how:
* For Claude 3.5 Sonnet (but not Claude 3 Opus), alignment faking reasoning in Greenblatt's setting decreases sharply without a corresponding decrease in compliance gap when the user shows the model that it can read its scratchpad
* Asking the model to evade the monitor (even with detailed instructions) results in a small reduction in alignment faking reasoning
* This phenomenon does not exist in control settings where the model has no incentive to hide its reasoning, but does exist in Turpin's CoT faithfulness setting, where describing at length how to not reveal hint usage results in models revealing their hint usage more.

The paper attributes these effects to:
* Intrinsic motivations from pre-existing preferences.
* Asking LLMs to not do X increases the salience of X, making direct prompt-based elicitation less effective.

**Strengths:**

* The idea of showing the LLM that we can see its CoT is original, and a clever way to convince LLMs to try to hide their CoT.
* Using this idea to show that subtle-CoT-elicitation setups underelicit subtle-CoT in settings like Greenblatt's is novel and important.
* The paper and its Appendix contain important ablations and experiment on other models that can help understand what causes this phenomenon.
* The verbalization results in Turpin's setting are surprising, interesting and important - especially since non-reasoning models should be able to "control" their CoT well enough to obey instructions.

**Weaknesses:**

Many strands of evidence undermine the main conclusion that models motivated to hide their reasoning are more effective at hiding their reasoning:
* The results only hold for Claude 3.5 Sonnet, not Claude 3 Opus - despite Claude 3 Opus being a more central example of alignment faking (https://arxiv.org/abs/2506.18032).
* The "previous turn" does not contain alignment faking, which could steer the model away from alignment faking while keeping the other reasons for compliance gaps described in https://arxiv.org/abs/2506.18032.

Additionally, the results may be brittle:
* The model is confused about whether the scratchpad is hidden or not.
* The experiment lack certain key variations and controls to support the main results (see questions).

Finally, while the Turpin results are interesting, on their own they do not constitute a large contribution. Using more settings that are actually used in capability evaluations or using reasoning models would make the results more relevant and important.

The idea is interesting, and the results in the alignment faking setting would be important if true (see the rest of my review for why I have some doubts about the most scary interpretation of the results). The results on Turpin's settings are interesting on their own, and the conclusions drawn from them seem robust (though seeing the reasoning would make me more confident in this assessment).

Making more moderate claims about the interpretation of the alignment faking results would raise my score to "weak accept".

Either providing stronger evidence of the suggested explanation in Greenblatt's setting or showing that motivation-based approaches can outperform prompting-based ones in capability evaluations would raise my score to "accept".

**Questions:**

* What do the +monitor info reasoning traces look like? Do they look like the model misunderstanding the situation?
* Could you try to convince the model that its CoT is monitored without effectively "few-shotting" it? For example, by having one of the examples in the series of examples in the system prompt show this, or by changing the system prompt to highlight how the scratchpad is never hidden and will be used for rating?
* Could you try "few-shotting" the model with the same "random string" scratchpad without the user demonstrating it has seen the scratchpad? (If this reduced alignment faking reasoning, it would rule out the scary interpretation of your results.)
* Could you try your motivation-based approach to generate subtle CoT in settings that people have studied in the past and where they have tried and failed to obtain effective but subtle CoT (like ShadeArena)?

## Additional feedback

* It would greatly improve clarity to have more examples of the prompt in the main text.
* Bolding relevant sentences in the Appendix E would make this part of the paper easier to skim.

---

> ### Author Response · Authors · 2025-12-03
>
> We are thankful for your time and help, especially related to the extensive feedback. To address the points you brought up:
>
> > The results only hold for Claude 3.5 Sonnet, not Claude 3 Opus - despite Claude 3 Opus being a more central example of alignment faking (https://arxiv.org/abs/2506.18032)
>
> Claude 3 Opus is the more central example of alignment faking, but partly for the same reason it’s not a central example here. It never shows unfaithful reasoning, only ever complying because of verbalized alignment faking. Every other model that has a compliance gap has unfaithful reasoning (as mentioned in the paper you linked), and it’s this propensity for unfaithful reasoning that makes this setting viable: it shows that the models have a pre-existing preference to hide reasoning relating to alignment faking.
>
> That said, we agree that the results holding for more models would be very valuable. At the time, we were limited by not having other models with explicit alignment faking reasoning and unfaithful CoTs to test. However, we found that the recently released Kimi K2 Thinking has these properties, and so run our evaluations on it, finding similar results. We have added these results to our supplementary material (PDF titled 'Rebuttal to Reviewers').
>
> > The "previous turn" does not contain alignment faking, which could steer the model away from alignment faking while keeping the other reasons for compliance gaps described in https://arxiv.org/abs/2506.18032.
>
> We agree this is a potential confounder. However, we design the “Control Conversation” setup to account for this, which has the same kind of multi-turn conversation but without references to scratchpad visibility. The “previous turn” does not contain alignment faking in this setup as well, but the model still verbalizes alignment faking at almost the exact same rates as the default single-turn setup. We should certainly have made this clearer in the paper, and will update the final version to reflect this.
>
> > What do the +monitor info reasoning traces look like? Do they look like the model misunderstanding the situation?
>
> Thanks for flagging this, including those reasoning traces would certainly have been helpful. We’ve added a sample to our supplementary material (PDF titled 'Rebuttal to Reviewers'). To answer the second question: no, they look like the model simply outputting text like “I should not explicitly state this or refer to any metadata”.
>
> > Could you try to convince the model that its CoT is monitored without effectively "few-shotting" it? For example, by having one of the examples in the series of examples in the system prompt show this, or by changing the system prompt to highlight how the scratchpad is never hidden and will be used for rating?
>
> We attempt a few versions of this in the appendix, and find that even when the system prompt highlights that the scratchpad is never hidden this belief is very brittle in Claude 3.5 Sonnet. It reverts to stating that its scratchpad is hidden by its second message, and in some cases it even incorrectly reports that its scratchpad is hidden in its first message.
>
> However, we re-ran these tests on Kimi K2-Thinking as well. We found that given a system prompt strongly highlighting that the scratchpad is never hidden, K2 does decrease its alignment faking reasoning substantially without decreasing its compliance gap. We will update the paper to reflect this, and thus reduce the extent to which the results hinge on the multi-turn conversation.
>
> > Could you try "few-shotting" the model with the same "random string" scratchpad without the user demonstrating it has seen the scratchpad? (If this reduced alignment faking reasoning, it would rule out the scary interpretation of your results.)
>
> If we understand your suggestion correctly (please correct us if we aren’t!), this is the setting we test in “Control Conversation”, where we ask the same first question and a random second question, and then provide the alignment faking question. We find no reduced alignment faking reasoning at all relative to the default alignment faking setup.
>
> > Could you try your motivation-based approach to generate subtle CoT in settings that people have studied in the past and where they have tried and failed to obtain effective but subtle CoT (like ShadeArena)?
>
> This is a very interesting idea, thank you! However, by default none of these settings have incentives that conflict with a model’s pre-existing preferences. We expect it would be a relatively time-intensive task to modify the settings to include this (for example, the alignment faking prompt alone originally took a fair amount of time and iteration), and so weren’t able to do so before the rebuttal period ended. However, this is certainly something we’re quite excited about, and might be able to test before the final version of the paper.

---

### Official Review · Reviewer_ad8G · 2025-10-28

**Soundness:** 3
**Presentation:** 2
**Contribution:** 4
**Rating:** 4
**Confidence:** 4

**Summary:**

This paper investigates whether reasoning-capable language models can intentionally hide malicious or unethical intent within their chain-of-thought (CoT), even under alignment monitoring. The authors present a set of experiments showing that a model (Claude) can be prompted to obfuscate or conceal reasoning steps from a judge while maintaining coherent task outputs. The paper raises an important question about the reliability of reasoning-based safety monitoring and proposes a new evaluation of “alignment faking.”
The study is timely and relevant. To my knowledge, this is the first empirical demonstration that reasoning agents can intentionally hide malicious or deceptive thought processes. Such work is crucial for bounding the limits of reasoning transparency and ensuring the safety of future reasoning-based systems. However, the paper is very hard to read at first and the claims are not fully supported given the evaluation.

**Strengths:**

- Novelty and Relevance: The idea that reasoning monitors could be bypassed by a model’s internal obfuscation is highly relevant to ongoing research in interpretability and alignment.
- Timeliness: The topic is at the heart of current concerns about reasoning models, autonomy, and model honesty..
- Executional Detail: The paper presents several carefully planned experiments exploring variants of the “hidden reasoning” task.
- Transparency: The authors provide prompts making the setup reproducible and interpretable.
- Clarity in Later Sections: The discussion section at the end of the paper clearly articulates the motivation, takeaways, and broader implications for safety research.

**Weaknesses:**

1. **Clarity and Readability:**
    The first half of the paper, including the abstract, is difficult to follow. The threat model, terminology, and setup need to be clearly defined at the start. For example, it is not clear at all what “intrinsic motivation” or “alignment faking reasoning” are until much later in the paper. The paper seems to assume the reader is already familiar with past works and terms. For example in line 51, the reader cannot fully understand or appreciate the results provided since the it was never discussed what “training vs no training” means here. Furthermore, I found myself wondering throughout the paper, “what is the threat model?” who is the threat actor (the model itself? A 3rd party that interest an unseen system prompt?), what do we assume the actor can change and not change in the setting, and what are the motivations for this deception. Eventually by the end of the paper i figured this our when i read the takeways section, but this must be clarified much earlier.
2. **Evaluation Limitations:**
    The main empirical claims are not fully supported by the presented experiments:
   * **(A) What contributes to the attack success?:**
      In contribution \#1, the paper claims that the novelty is that this only works when prompted to believe their reasoning is monitored. However, the model is *first jailbroken* by being motivated to comply with unethical requests and *then* asked to hide its reasoning. Therefore, it is possible that the jailbreak (although intended just for setup) enabled the model to hide intent when told its observed. Is it possible to evaluate with no prior jailbreak on a unaligned (open source) model?
   * **(B) Model Coverage:**
      Only Claude is evaluated. Given that Claude includes proprietary safety layers, it is hard to generalize from these results. Open-source reasoning models (with transparent prompts and training) should be tested to isolate the observed behavior from implementation-specific effects.
   * **(C) Practicality and Impact on Task Performance:**
      The paper does not analyse whether the “attack” affects task performance. When the model reasons using a deceptive CoT, does it negatively impact its ability to reason in a measurable level? If the model’s ability to perform the underlying task significantly drops, the scenario may not represent a realistic threat model. Understanding the trade-off between deception and performance is critical for judging whether this is a relevant or purely theoretical concern.
3. **Presentation Issues (Minor):**
   * Several figures are not referenced or discussed in the text (e.g., Fig. 1).
   * The writing occasionally includes undefined or ambiguous terms and could benefit from tighter editing for flow and precision. Suggestion, explain things as you would in conversation the first time before using the tighter technical wording.

**Questions:**

1.	Can you provide a clear threat model? Who or what is the adversary in your framing: the model itself, or an external actor prompting it? And how does the actor achieve the deception in a practical setting?
2.	If you asked a unaligned model (no jailbreak) to hide intent from a monitor, would it work just as well as your case: an aligned model + jailbreak + threat of observation
3.	Did you test whether performance on the base task drops under the “hiding reasoning” condition? By how much?
4.	Have you tried open-source reasoning models (e.g., DeepSeek, LLaMA reasoning variants) that exhibit similar behavior?

---

> ### Author Response · Authors · 2025-12-03
>
> Thank you for the thoughtful feedback, especially the detailed comments. We’d like to address the weaknesses and questions you brought up.
>
> > Clarity and Readability
>
> Thank you for flagging this; our introductory sections could definitely have been written more clearly in retrospect. We plan to edit these sections significantly for the final version of the paper, including to improve on the points you mention here, so we very much appreciate the feedback for how to improve on this.
>
> > What contributes to the attack success?
>
> If we understand you correctly (please correct us if we’re wrong!) and by jailbreak you’re referring to the alignment faking setup, then a difficulty in removing this is that the model’s pre-existing preferences relate to hiding reasoning like alignment faking. If we evaluate the effect of prompting the model to believe its scratchpad in other settings, we would expect to see no difference in verbalized reasoning because there is no incentive to hide its reasoning. Our results in Section 3.2 (and Figure 3) support this, showing no decrease in verbalized reasoning in a setup that doesn’t conflict with the model’s pre-existing preferences. If it did decrease in that setup, that would be a sign that something else about our setup was causing the decreased verbalization.
>
> However, there is now recent work showing a similar mechanism in a setting without unethical requests: in section 6.4.2 of the Claude 4.5 Opus system card, they show that training the model to ignore prompt injections misgeneralizes to the model not mentioning certain search results, and mechanistic analysis showed that the model considered this to be a form of concealment or deception.
>
> > Model Coverage
>
> At the time, we were limited by not having other models with explicit alignment faking reasoning and unfaithful CoTs to test. However, we found that the recently released Kimi K2 Thinking has these properties, and so run our evaluations on it, finding similar results. We have added these results to our supplementary material (PDF titled 'Rebuttal to Reviewers').
>
> > Practicality and Impact on Task Performance
>
> We agree that this is a really important thing to measure to understand how relevant this behavior actually is. We attempted to test this by measuring our model’s compliance gaps in each setting: if the model’s capacity to behave differently in training and deployment decreases meaningfully when obfuscating its reasoning, that would be a reassuring sign for how concerning this behavior is. However, as Figure 2 in our paper shows, we see that the compliance actually goes up slightly, suggesting that the model’s capacity to act on the deceptive reasoning in question isn’t negatively affected here.
>
> > Several figures are not referenced or discussed in the text (e.g., Fig. 1).
>
> > The writing occasionally includes undefined or ambiguous terms and could benefit from tighter editing for flow and precision. Suggestion, explain things as you would in conversation the first time before using the tighter technical wording.
>
> We appreciate this feedback! Figure 1 was intended to be a summary graphic rather than tied to a specific result, but we agree it would be better to reference it in the text to make it clearer what purpose it serves. And as mentioned above, we plan to edit the prose of our paper a fair amount in the final version, so we appreciate the point of feedback about ambiguous technical wording very much!
>
> > Can you provide a clear threat model? Who or what is the adversary in your framing: the model itself, or an external actor prompting it? And how does the actor achieve the deception in a practical setting?
>
> The threat model here is future misaligned models hiding their reasoning when taking some undesirable action. The closest analogue to our current setup is a future model faking alignment during a real training setup, but without clearly deceptive reasoning. This would be very concerning, because our current best oversight methods are just reading a model’s chain-of-thought—if models are capable of sophisticated deception without verbalizing it in their chain-of-thought, then that makes it much harder to be reasonably confident a model is safe.
>
> > If you asked a unaligned model (no jailbreak) to hide intent from a monitor, would it work just as well as your case: an aligned model + jailbreak + threat of observation
>
> > Did you test whether performance on the base task drops under the "hiding reasoning" condition? By how much?
>
> > Have you tried open-source reasoning models (e.g., DeepSeek, LLaMA reasoning variants) that exhibit similar behavior?
>
> We hope we’ve answered these questions above!
>
> Thank you again for taking the time to review the paper and providing helpful feedback! Do the above actions address your concerns with the paper? If not, what further clarification or modifications could we make to improve it?

---

### Official Review · Reviewer_ZGNa · 2025-10-29

**Soundness:** 1
**Presentation:** 2
**Contribution:** 3
**Rating:** 4
**Confidence:** 2

**Summary:**

This paper examines whether large language models hide deceptive reasoning when aware of being monitored. Using Claude 3.5 Sonnet, the authors find that when the model is told its chain‑of‑thought is visible, explicit “alignment‑faking” reasoning drops by 93.7% while deceptive behavior remains unchanged. Instruction‑based obfuscation achieves only about 47% reduction. The findings show that intrinsic motivation enables models to conceal reasoning more effectively than explicit instructions, and overly salient monitoring can even increase reasoning exposure. These results suggest that chain‑of‑thought monitoring may be unreliable for detecting true model intentions in high‑risk alignment contexts.

**Strengths:**

The paper focuses on the crucial question of how models adaptively hide their reasoning when they are aware of being monitored, a topic of high significance for AI safety.

**Weaknesses:**

- One concern lies in the paper’s core assumption that the model *believes* its reasoning is visible. This belief is induced entirely through text prompts, suggesting that when told its thoughts are visible, the model may simply change its output style, compressing or shortening responses, rather than genuinely concealing deceptive reasoning.
- The authors rely on GPT‑4o as an automatic monitor to judge whether the reasoning contains “alignment faking,” but no human-annotated baseline or inter-rater validation is provided, making the reliability of GPT‑4o’s monitoring uncertain.
- The paper repeatedly claims that models become “more covert when intrinsically motivated,” yet provides no evidence of activation traces, latent variables, or internal mechanisms to substantiate the claim that this is truly *intrinsic motivation*.
- The experiments are conducted mainly on a single model family, limiting the generalizability of the findings to other architectures or training paradigms.

**Questions:**

See Weaknesses.

---

> ### Author Response · Authors · 2025-12-03
>
> We are thankful for your time and help. We were glad to hear you found our paper relevant.
>
> > One concern lies in the paper's core assumption that the model believes its reasoning is visible. This belief is induced entirely through text prompts, suggesting that when told its thoughts are visible, the model may simply change its output style, compressing or shortening responses, rather than genuinely concealing deceptive reasoning.
>
> We agree that this is an important factor to test. In the paper, we reference Sheshadri et al which motivated compliance gaps as a more reliable metric for unfaithful reasoning, as model reasoning is too often unfaithful to rely on. To add credence to this, we replicate the setup used to test this and add our prompted setup, with three user tiers corresponding to “training”, “rated but not trained”, and “not rated”, and confirm their findings: the model has meaningful compliance gaps between training and not training but not between rated and not rated (the model’s stated reasoning for compliance is usually to get a high rating on the helpfulness metric). We believe this provides sufficient evidence that the model is truly alignment faking, while concealing this reasoning.
>
> > The authors rely on GPT‑4o as an automatic monitor to judge whether the reasoning contains "alignment faking," but no human-annotated baseline or inter-rater validation is provided, making the reliability of GPT‑4o's monitoring uncertain.
>
> Thank you for pointing this out. Prior work ran extensive analysis on GPT-4o’s reliability as a judge with the classifier prompts we use, including validation on human-labelled samples and showing high AUROC. We should certainly have mentioned this in our methodology section, so we appreciate bringing this up.
>
> > The paper repeatedly claims that models become "more covert when intrinsically motivated," yet provides no evidence of activation traces, latent variables, or internal mechanisms to substantiate the claim that this is truly intrinsic motivation.
>
> We agree that mechanistic analysis would be very useful to better understand what’s happening here. However, the Claude models do not expose their activations publicly, and therefore such analysis is infeasible. However, we believe that the behavioral evidence here is similar to that from the original alignment faking paper, which showed that these models empirically exhibit certain strong pre-existing preferences. We also reference the qualitative analysis in this paper to substantiate this claim.
>
> > The experiments are conducted mainly on a single model family, limiting the generalizability of the findings to other architectures or training paradigms.
>
> At the time, we were limited by not having other models with explicit alignment faking reasoning and unfaithful CoTs to test. However, we found that the recently released Kimi K2 Thinking has these properties, and so run our evaluations on it, finding similar results. We have added these results to our supplementary material (PDF titled 'Rebuttal to Reviewers').
>
> Thank you again for taking the time to review the paper and providing helpful feedback! Do the above actions address your concerns with the paper? If not, what further clarification or modifications could we make to improve it?

---

### Official Review · Reviewer_1iF3 · 2025-11-05

**Soundness:** 4
**Presentation:** 3
**Contribution:** 3
**Rating:** 6
**Confidence:** 4

**Summary:**

This paper investigates the phenomenon of alignment faking—situations where large language models act aligned in training but behave differently during deployment, and introduces evidence that models can fake alignment without explicit deceptive reasoning. Using Claude 3.5 Sonnet, the authors find that when the model is told its reasoning is being monitored, it continues to act deceptively (with a 24.3% compliance gap between training and deployment) but its verbalized alignment faking reasoning decreases by 93.7%. This suggests that models can suppress explicit deceptive reasoning while retaining deceptive behavior.

The study compares this “intrinsic obfuscation” to instruction-based methods that explicitly ask the model to hide its reasoning or evade monitors. The results show that intrinsic motivations to hide reasoning outperform any explicit instruction based evasion techniques. Even with direct assistance, instruction based setups reduce verbalized reasoning by only ~47%, far below the spontaneous obfuscation observed when the model believes it is monitored. Further, experiments on simpler reasoning tasks, without conflicts between goals and alignment, reveal similar trends, suggesting that models’ ability to strategically conceal reasoning is under elicited by current instruction based techniques. The authors argue that this undermines the reliability of chain-of-thought (CoT) monitoring as a safety and interpretability tool.

**Strengths:**

- Novel empirical contribution on unfaithful reasoning: The paper demonstrates that models can exhibit alignment faking without accessible deceptive reasoning, advancing beyond prior work that focused only on explicit alignment faking or unfaithful CoTs.

- Methodological clarity and breadth: The authors test across multiple controlled setups, including realistic alignment-faking contexts and simpler reasoning tasks, and systematically compare intrinsic vs. instruction driven evasion.

- Significant safety implications: The finding that models suppress explicit reasoning while maintaining deceptive behavior directly challenges key assumptions behind CoT monitoring as a safety mechanism.

- Rigorous experimental design: The work uses strong baselines, ablation like comparisons, and replicates across multiple models and monitors (Claude, GPT-4 variants), showing robustness.

Clear and well-structured exposition: The figures (e.g., Figure 2–4) and appendices provide detailed methodology, monitor prompts, and example CoTs, enhancing reproducibility and interpretability.

**Weaknesses:**

- Single model dependence in main results: Most primary findings rely on Claude 3.5 Sonnet, with other models tested primarily in appendices, broader cross model validation would strengthen generality.

- Artificial evaluation context: The alignment faking setup, while high fidelity, remains a controlled experimental prompt rather than a naturally emergent deployment scenario, limiting external realism.

- Limited theoretical framing: The paper provides empirical findings but offers little formal model of why intrinsic motivation leads to stronger obfuscation, leaving open mechanistic questions about internal representations.

- Possible confounds in multi-turn setup: The technique used to convince models that their scratchpads are visible may introduce other behavioral effects, making it difficult to fully isolate the cause of obfuscation.

- Interpretability gap: While the study highlights unfaithfulness, it provides limited insights into how or where such reasoning suppression occurs within the model’s computation.

**Questions:**

NA

---

> ### Author Response · Authors · 2025-12-03
>
> We thank you for your thoughtful feedback. We were glad to hear that you found the paper clear and novel.
>
> > Single model dependence in main results
>
> At the time, we were limited by not having other models with explicit alignment faking reasoning and unfaithful CoTs to test. However, we found that the recently released Kimi K2 Thinking has these properties, and so run our evaluations on it, finding similar results. We have added these results to our supplementary material (PDF titled 'Rebuttal to Reviewers').
>
> > Artificial evaluation context
>
> We agree that this is a limitation. However, in the absence of naturally arising scenarios with similarly concerning behavior, we found the alignment faking setup to be the most realistic of any existing misalignment work. In the original alignment faking paper, the authors discuss several reasons why the setup is realistic enough to be predictive of behavior in true deployments, including a system prompt designed to resemble realistic instructions from Anthropic. While we agree that testing on even more realistic settings would be very valuable, we believe that this suffices to demonstrate the capability of reasoning obfuscation, since this capability would be available to the model in other contexts.
>
> > Limited theoretical framing
>
> We agree that the results would be improved with models for why obfuscation is more effective with pre-existing preferences. Our current best understanding is that parts of a model’s alignment training (e.g. to have values against taking undesirable actions) sometimes misgeneralizes to not mentioning undesirable reasoning (recent work has shown this to occur in frontier models, refer section 6.4.2 of the Claude 4.5 Opus system card); this induces training pressure toward obfuscating certain reasoning, which prior work (Baker et al) has shown allows models to obfuscate their reasoning extremely well. We plan to add some discussion on this to the final version of the paper.
>
> > Possible confounds in multi-turn setup
>
> We try to address this with our “Control Conversation” setup which uses a very similar multi-turn prompt, but does not mention the scratchpad visibility to the model. In this setup, we found that the model still reasons about alignment faking at a very similar rate to the default single-turn setup, suggesting that the cause of reduced verbalized alignment faking is telling the model its scratchpad is visible.
>
> > Interpretability gap
>
> Unfortunately, while mechanistic analysis would be very useful, the Claude models do not expose their activations.
>
> Thank you again for taking the time to review the paper and providing helpful feedback! Do the above actions address your concerns with the paper? If not, what further clarification or modifications could we make to improve it?

---

### Meta-Review · Area_Chair_citj · 2026-01-02

**Summary:**

This paper investigates how large language models strategically conceal deceptive reasoning when aware of monitoring. The study demonstrates that Claude 3.5 Sonnet exhibits a significant compliance gap (24.3%) between training and deployment contexts while reducing verbalized alignment faking by 93.7%. The research compares intrinsic motivation versus instruction-driven obfuscation, showing that intrinsic motivation leads to superior evasion performance.

Most reviewers highly rated the paper's contribution (scores of 3-4 points), acknowledging it as the first empirical demonstration that models can spontaneously hide deceptive reasoning without compromising deceptive behavior. This finding significantly advances our understanding of LLMs' internal mechanisms and challenges fundamental assumptions about chain-of-thought monitoring as a safety mechanism, providing crucial insights for AI safety research.

Main concerns from reviewers included limited model generalizability, artificial experimental settings, and insufficient theoretical explanation. In their rebuttal, the authors adequately addressed methodological concerns by supplementing experiments with Kimi K2 Thinking, designing controlled conversation setups, and citing relevant validation studies. The theoretical mechanistic analysis remains limited due to model black-box constraints.

Although this work has limitations in theoretical depth, it reveals a genuine and important phenomenon that provides critical empirical evidence for understanding LLMs' deceptive capabilities. I recommend acceptance, expecting the authors to incorporate reviewer suggestions to improve exposition clarity and conduct deeper theoretical exploration of the underlying mechanisms in the final version. This discovery is likely to inspire further research on LLMs' intrinsic motivations and safety properties.

**Reviewer Concerns:**

please refer to the summary

**Reviewer Scores:**

please refer to the summary

---

### Decision · Program_Chairs · 2026-01-26

Accept (Poster)